# Tissue-adjusted pathway analysis of cancer (TPAC): A novel approach for quantifying tumor-specific gene set dysregulation relative to normal tissue

H. Robert Frost●*

Department of Biomedical Data Science, Geisel School of Medicine, Dartmouth College, Hanover, New Hampshire, United States of America

* rob.frost@dartmouth.edu

## Abstract

We describe a novel single sample gene set testing method for cancer transcriptomics data named tissue-adjusted pathway analysis of cancer (TPAC). The TPAC method leverages information about the normal tissue-specificity of human genes to compute a robust multi-variate distance score that quantifies gene set dysregulation in each profiled tumor. Because the null distribution of the TPAC scores has an accurate gamma approximation, both population and sample-level inference is supported. As we demonstrate through an analysis of gene expression data for 21 solid human cancers from The Cancer Genome Atlas (TCGA) and associated normal tissue expression data from the Human Protein Atlas (HPA), TPAC gene set scores are more strongly associated with patient prognosis than the scores generated by existing single sample gene set testing methods.

**Data Availability Statement:** Cancer and normal tissue gene expression data was accessed from The Cancer Genome Atlas (TCGA) and Human Protein Atlas (HPA). For TCGA data, the PANCAN

## Author summary

Cancer biology is highly tissue-specific: most cancer-driving somatic alterations occur in just a limited number of tissues and inherited mutations frequently have a tissue-specific functional impact. To leverage the associations between gene activity in normal and malignant tissue for gene set testing, we have developed a new single sample gene set testing method for tumor-derived transcriptomics data named TPAC (tissue-adjusted pathway analysis of cancer). The TPAC method uses normal tissue-specificity to quantifies gene set dysregulation in each profiled tumor. Importantly, we found that TPAC gene set scores are more strongly associated with patient prognosis than the scores generated by existing single sample gene set testing methods.

## Introduction

Cancer develops when somatic alterations disrupt pathways associated with genome maintenance, cell proliferation, or cell survival to give a relative growth advantage to the impacted

RNA-seq and phenotype data contained in the files 'GDCPANCAN.htseq fpkm.tsv.gz' and 'GDCPANCAN.GCD phenotype.tsv' was downloaded from the GCD Data Portal (https://gdc.cancer.gov/). Alternative cancer outcomes (e.g., progression-free interval) were accessed from the TCGA Pan-Cancer Clinical Data Resource (https://gdc.cancer.gov/about-data/publications/PanCan-Clinical-2018). Note that an equivalent FPKM-normalized version of the TCGA RNA-seq data can be access from the HPA project at https://www.proteinatlas.org/download/rna cancer sample.tsv.zip. For HPA data, the HPA staff provided normal tissue gene expression data in the file 'HPA.normal.FPKM.GDCpipeline.csv'; this data was especially normalized by the HPA group as FPKM using a pipeline similar to that employed by GDC for the TCGA data (this data was generated for the 'Human Pathology Atlas' paper and can be accessed at https://hrfrost.host.dartmouth.edu/TPAC). The Hallmark collection gene sets were downloaded from version 7.2 of the Molecular Signatures Database (MSigDB) (as downloaded from http://software.broadinstitute.org/gsea/downloads.jsp). The TPAC R package and imported TPACData package are available on CRAN at https://cran.r-project.org/web/packages/TPAC and https://cran.r-project.org/web/packages/TPACData.

**Funding:** HRF received funding from National Institutes of Health grants R21CA253408, R35GM146586, P20GM130454 and P30CA023108. The funders had no role in study design, data collection and analysis, decision to publish, or preparation of the manuscript.

**Competing interests:** The authors have declared that no competing interests exist.

cell population [1]. Given the central role of pathway dysregulation in tumor initiation, growth and metastatic spread, basic and translational cancer research is often focused on pathway-related questions, e.g., How do somatic alterations cause dysregulation of specific pathways? What is the impact of pathway dysregulation on patient prognosis? What types of therapeutic agents can restore normal pathway function? One common approach for answering these questions involves the analysis of tumor-derived genomic data using gene set testing, or pathway analysis, methods. Gene set testing is a hypothesis aggregation technique that evaluates statistics computed on functionally related groups of genes, e.g., the sets maintained in collections like the Molecular Signatures Database (MSigDB) [2]. By focusing on gene sets, rather than individual genes, gene set testing can significantly improve power, interpretation and replication [3–6]. It should be noted that this paper uses the terms pathway and gene set interchangably to refer to an unordered group of genes, i.e., no knowledge is assumed regarding the regulatory or other relationships between the genes in set/pathway. This usage is in contrast to a distinction that is often made between these terms with 'gene set' representing an unordered groups of genes, e.g., Gene Ontology term [7], and 'pathway' a model that captures the regulatory relationships among a set of genes, e.g, Reactome pathways [8].

Although the pathways most commonly impacted in cancer have been identified [1] and progress has been made developing cancer-specific pathway analysis methods that can integrate gene expression and somatic alteration data [9–13], current approaches leverage just tumor-specific genomic data and do not take into account gene activity in the associated normal tissue. This is an important limitation given the strong association between gene activity in normal tissues and the pathophysiology of cancers originating in those tissues. Cancer biology is highly tissue and cell type-specific [14–18] with most driving somatic alterations either occuring in only a small number of cancer types, e.g., KRAS mutations in pancreatic, lung and colorectal cancers [19], or having a functional impact that varies between impacted tissues, e.g., germline BRCA1/BRCA2 mutations that only drive cancer in estrogen-sensitive tissues [20]. As we explored in a recent paper [21], this tissue-specificity means that the pattern of gene activity (as quantified by mRNA expression) in normal tissues carries important information regarding the biology of the associated cancer types. In that paper, we demonstrated through an analysis of solid tumor data from The Cancer Genome Atlas (TCGA) [22] and normal tissue data from the Human Protein Atla (HPA) [23] that the association between tissue-specific and cancer-specific expression values, i.e., the ratio of gene expression in a specific tissue or cancer relative to the mean across multiple tissues or cancers, can be used to improve survival analysis, the comparative analysis of distinct cancer types, and the analysis of cancer/normal tissue pairs. Specifically, we found that normal tissue-specific genes (i.e., genes that are highly-expressed in a given normal tissue relative to other tissue types) are typically down-regulated in the associated cancer (see the $\rho_{cancer/norm}$ column in Table 1). We also found that elevated expression of tissue-specific genes in a tumor is associated with improved cancer prognosis (see the $\rho_{surv}$ column in Table 1). As detailed in that paper [21], these associations can be leveraged via gene filtering/weighting approaches to improve the statistical power for both cancer survivaland cancer vs. normal differential expression analyses, and to identify transcriptomic differences between different cancers that are independent of the associated normal tissues.

While a number of gene test testing approaches have been explored that account for tissue or cell type-specificity [24–27], the primary goal of these techniques was the identification of tissue-specific biological processes. None of these prior methods specifically focuses on cancer or uses the association between normal tissue-specificity and gene activity within dysplastic tissue. Given the lack of gene set testing methods that can leverage the association between normal tissue-specificity and cancer biology, we developed a new single sample gene set testing

**Table 1. The 21 TCGA cancer types and corresponding HPA normal tissues supported by TPAC.** The $\rho_{cancer/norm}$ column holds the Spearman rank correlation between normal tissue-specific gene weights (the log fold-change of the mean expression in the normal tissue to the mean in all normal tissues) and the log fold-change of the mean expression in the cancer type to mean expression in the normal tissue. The $\rho_{surv}$ column holds the Spearman rank correlation between normal tissue-specific gene weights and the signed log of the p-value from a Kaplan-Meir test of the association between gene expression and cancer survival as computed by Uhlen et al. [23]), which is computed as -log(p-value) for favorable genes and log(p-value) for unfavorable genes. The contents of this table are a synthesis of information from tables 2 and 3 in Frost [21].

| TCGA disease code | Cancer type | HPA tissue | $\rho_{cancer/norm}$ | $\rho_{surv}$ |
|---|---|---|---|---|
| BLCA | Bladder Urothelial Carcinoma | urinary bladder | -0.299 | -0.0229 |
| BRCA | Breast Invasive Carcinoma | breast | -0.437 | -0.0993 |
| CESC | Cervical Squamous Cell Carcinoma | cervix, uterine | -0.432 | -0.105 |
| COAD | Colon Adenocarcinoma | colon | -0.057 | 0.304 |
| GBM | Glioblastoma Multiforme | cerebral cortex | -0.446 | 0.0764 |
| HNSC | Head and Neck Squamous Cell Carcinoma | tonsil | -0.48 | 0.11 |
| KICH | Kidney Chromophobe | kidney | -0.232 | 0.454 |
| KIRC | Kidney Renal Clear Cell Carcinoma | kidney | -0.436 | 0.454 |
| KIRP | Kidney Renal Papillary Cell Carcinoma | kidney | -0.311 | 0.454 |
| LIHC | Liver Hepatocellular Carcinoma | liver | -0.358 | 0.35 |
| LUAD | Lung Adenocarcinoma | lung | -0.363 | 0.073 |
| LUSC | Lung Squamous Cell Carcinoma | lung | -0.406 | 0.073 |
| OV | Ovarian Serous Cystadenocarcinoma | ovary | -0.546 | -0.0348 |
| PAAD | Pancreatic Adenocarcinoma | pancreas | -0.643 | 0.148 |
| PRAD | Prostate Adenocarcinoma | prostate | -0.179 | -0.0397 |
| READ | Rectum Adenocarcinoma | rectum | -0.208 | 0.349 |
| SKCM | Skin Cutaneous Melanoma | skin | -0.444 | -0.0707 |
| STAD | Stomach Adenocarcinoma | stomach | -0.258 | 0.344 |
| TGCT | Testicular Germ Cell Tumors | testis | -0.636 | 0.256 |
| THCA | Thyroid Carcinoma | thyroid gland | -0.441 | -0.0851 |
| UCEC | Uterine Corpus Endometrial Carcinoma | endometrium | -0.45 | 0.0276 |

method for tumor-derived transcriptomics data named TPAC (tissue-adjusted pathway analysis of cancer). The TPAC method uses information about the normal tissue-specificity of human genes to compute a robust multivariate distance score that quantifies pathway dysregulation in each profiled tumor. TPAC currently supports the 21 solid tumor types listed in Table 1. As we demonstrate through an analysis of TCGA gene expression data, TPAC gene set scores are more strongly associated with both patient prognosis and tumor stage than the scores generated by existing single sample gene set testing methods. In the remainder of this paper, we detail the TPAC method in and evaluate the performance of TPAC relative to existing single sample methods.

## Materials and methods

### Data sources

The findings detailed in this paper are based on bulk RNA-seq and clinical data from The Cancer Genome Atlas (TCGA) [22] for 21 human solid cancers and bulk RNA-seq data from the Human Protein Atlas (HPA) [28] for the associated 18 normal human tissues. These cancer types and normal tissues are listed in Table 1. Alternative cancer outcomes (e.g., progression-free interval) were accessed from the TCGA Pan-Cancer Clinical Data Resource (TCGA-CDR) [29]. The Hallmark gene set definitions were accessed from version 7.2 of the Molecular Signatures Database (MSigDB) [2]. As mentioned above regarding the distinction between gene sets and pathways, it is worth noting that MSigDB only includes unordered gene sets and, for some

collections, e.g., Gene Ontology [30] and KEGG [31], significant filtering is performed. Researchers interested in leveraging the TPAC method should therefore explore not only MSigDB collections in addition to the Hallmark sets but the unmodified versions of collections such as the Gene Ontology, KEGG and Reactome [8]. Additional details on the data sources used in this paper can be found in the data availability section and the Methods section in S1 Text.

## TPAC method

The TPAC method generates single sample gene set scores from tumor gene expression data using a modified version of the classic Mahalanobis multivariate distance measure [32]. TPAC takes four inputs:

- **X**: $n \times p$ matrix that holds the normalized expression measurements for $p$ genes in $n$ tumors all of the same type (e.g., normalized RNA-seq data from TCGA). For the current implementation of TPAC, the tumor type is limited to one of the 21 solid cancers listed in Table 1. Genes with 0 variance are removed.

- **t**: a length $p$ vector that holds the mean gene expression values (as quantified by the HPA) for the normal tissue associated with the cancer type whose data is held in **X**.

- $\bar{\mathbf{t}}$: a length $p$ vector that holds the average of the mean gene expression values across all 18 normal tissues in Table 1.

- **A**: $m \times p$ matrix that represents the annotation of $p$ genes to $m$ gene sets as defined by a collection from a repository like the Molecular Signatures Database (MSigDB) [2] ($a_{i,j} = 1$ if gene $j$ belongs to gene set $i$).

Given **X**, **t**, $\bar{\mathbf{t}}$, and **A**, TPAC computes $n \times m$ matrices **S**, $\mathbf{S}^+$, and $\mathbf{S}^-$. These matrices hold single sample scores for each of the $m$ gene sets defined in **A** and $n$ tumors captured in **X**. These single sample gene set scores are computed as follows:

1. **Compute normal tissue-specificity**: Let the length $p$ vector $\mathbf{t}^*$ hold values representing the normal tissue-specificity of the $p$ genes in **X**. These tissue-specific values are computed as $\mathbf{t}^* = \mathbf{t}/\bar{\mathbf{t}}$, i.e., the fold-change in mean expression between the normal tissue associated with the target cancer type and the average in all 18 normal tissues listed in Table 1.

2. **Compute expression deviation between each tumor and associated normal tissue**: Let $\Delta$ hold the differences between the expression values in **X** and the normal tissue means in **t**, i.e, row $i$ in $\Delta$ is computed by subtracting **t** from row $i$ of **X**. Define two versions of $\Delta$, $\Delta^+$ and $\Delta^-$, that capture just the positive or just the negative deviations. Specifically, element $\delta_{i,j}^+$ of $\Delta^+$ is set to element $\delta_{i,j}$ of $\Delta$ if $\delta_{i,j} \geq 0$, otherwise, it is set to 0. Similarly, element $\delta_{i,j}^-$ of $\Delta^-$ is set to element $\delta_{i,j}$ of $\Delta$ if $\delta_{i,j} < 0$, otherwise, it is set to 0.

3. **Compute weighted sample covariance matrices**: Two weighted versions of the unbiased sample covariance matrix, $\hat{\Sigma}^+$ and $\hat{\Sigma}^-$, are computed that adjust the sample variance according to normal tissue-specificity. Specifically, diagonal element $\hat{\sigma}_{i,i}^+$ of $\hat{\Sigma}^+$ is set to the sample variance of gene $i$ as computed on **X** multiplied by $t_i^*$ (the tissue-specificity value for gene $i$). Similarly, diagonal element $\hat{\sigma}_{i,i}^-$ of $\hat{\Sigma}^-$ is set to the sample variance of gene $i$ divided by $t_i^*$. All off-diagonal elements in $\hat{\Sigma}^+$ and $\hat{\Sigma}^-$ are set to 0. Table 2 captures the impact of this weighting on the sample variance for gene $i$ (i.e., $\hat{\sigma}_i$), e.g., the variance is inflated for genes that are up-regulated in the normal tissue relative to other tissues (i.e., $t_i^* > 1$) and have

**Table 2. Impact of tissue-specific weighting on sample variance for gene $i$.**

|  | $t_i^* \geq 1$ | $t_i^* < 1$ |
|---|---|---|
| $\delta_{i,j} \geq 0$ | $\hat{\sigma}_i \uparrow$ | $\hat{\sigma}_i \downarrow$ |
| $\delta_{i,j} < 0$ | $\hat{\sigma}_i \downarrow$ | $\hat{\sigma}_i \uparrow$ |

elevated expression in the tumor relative to normal tissue (i.e., $\delta_{i,j} \geq 0$). The impact of these variance changes is discussed in more detail below.

4. **Compute modified Mahalanobis distances for positive expression deviations**: Let $\mathbf{M}^+$ be an $n \times m$ matrix of squared values of modified Mahalanobis distances. Each column $k$ of $\mathbf{M}^+$, which holds the positive component of the sample-specific squared distances for gene set $k$, is calculated as:

$$\mathbf{M}^+[, k] = \mathrm{diag}(\boldsymbol{\Delta}_k^+ (\hat{\boldsymbol{\Sigma}}_k^+)^{-1} (\boldsymbol{\Delta}_k^+)^T) \tag{1}$$

where $g$ is the size of gene set $k$, $\boldsymbol{\Delta}_k^+$ is a $n \times g$ matrix containing the $g$ columns of $\boldsymbol{\Delta}^+$ corresponding to the members of set $k$, and $\hat{\boldsymbol{\Sigma}}_k^+$ is a $g \times g$ matrix containing the $g$ rows and columns of $\hat{\boldsymbol{\Sigma}}^+$ corresponding to the members of set $k$.

5. **Compute modified Mahalanobis distances for negative expression deviations**: Let $\mathbf{M}^-$ be an $n \times m$ matrix of squared values of modified Mahalanobis distances. Similar to $\mathbf{M}^+$, each column $k$ of $\mathbf{M}^-$, which holds the negative component of the sample-specific squared distances for gene set $k$, is calculated as:

$$\mathbf{M}^-[, k] = \mathrm{diag}(\boldsymbol{\Delta}_k^- (\hat{\boldsymbol{\Sigma}}_k^-)^{-1} (\boldsymbol{\Delta}_k^-)^T) \tag{2}$$

where $g$ is the size of gene set $k$, $\boldsymbol{\Delta}_k^-$ is a $n \times g$ matrix containing the $g$ columns of $\boldsymbol{\Delta}^-$ corresponding to the members of set $k$, and $\hat{\boldsymbol{\Sigma}}_k^-$ is a $g \times g$ matrix containing the $g$ rows and columns of $\hat{\boldsymbol{\Sigma}}^-$ corresponding to the members of set $k$.

6. **Compute modified Mahalanobis distances for positive and negative expression deviations**: Let $\mathbf{M}$ be an $n \times m$ matrix of squared values of modified Mahalanobis distances that capture both positive and negative expression deviations from the associated normal tissue. These total squared distances are simply computed as the sum of the positive and negative distances:

$$\mathbf{M} = \mathbf{M}^+ + \mathbf{M}^- \tag{3}$$

7. **Compute modified Mahalanobis distances on permuted $\mathbf{X}$**: To capture the distribution of the squared modified Mahalanobis distances under the $H_0$ that the expression values in $\mathbf{X}$ are uncorrelated with no mean difference between samples, the $\mathbf{M}$, $\mathbf{M}^+$, and $\mathbf{M}^-$ matrices are recomputed on a version of $\mathbf{X}$ where the row labels of each column are randomly permuted. Let $\mathbf{X_p}$ represent the row-permuted version of $\mathbf{X}$ and let $\mathbf{M}_p$, $\mathbf{M}_p^+$, and $\mathbf{M}_p^-$ that hold the squared modified Mahalanobis distances computed on $\mathbf{X_p}$ according to (1), (2), or (3).

8. **Fit gamma distribution to columns of $\mathbf{M}_p$, $\mathbf{M}_p^+$, and $\mathbf{M}_p^-$**: A separate gamma distribution is fit using the method of maximum likelihood (as implemented by the *fitdistr()* function in the MASS R package [33]) to the non-zero elements in each column of $\mathbf{M}_p$, $\mathbf{M}_p^+$, and $\mathbf{M}_p^-$.

Let $\hat{\alpha}_k$ and $\hat{\beta}_k, k \in 1, \ldots, m$ represent the gamma shape and rate parameters estimated for

gene set $k$ on $\mathbf{M}_p$ using this procedure. For $\mathbf{M}_p^+$ and $\mathbf{M}_p^-$, these estimated parameters are $\hat{\alpha}_k^+$, $\hat{\beta}_k^+$, $\hat{\alpha}_k^-$, and $\hat{\beta}_k^-$.

9. **Use gamma cumulative distribution function (CDF) to compute tumor-specific gene set scores**: The tumor-specific gene set scores are set to the gamma CDF value for each element of $\mathbf{M}$, $\mathbf{M}^+$, and $\mathbf{M}^-$. Specifically, each column $k$ of $\mathbf{S}$, which holds the tumor-specific scores for gene set $k$, is calculated as:

$$\mathbf{S}[,k] = F_{\gamma(\hat{\alpha}_k, \hat{\beta}_k)}(\mathbf{M}_p[,k]) \qquad (4)$$

where $F_{\gamma(\hat{\alpha}_k, \hat{\beta}_k)}()$ is the CDF for the gamma distribution with shape $\hat{\alpha}_k$ and rate $\hat{\beta}_k$. Under the $H_0$ of uncorrelated expression, valid p-values can be generated by subtracting the elements of $\mathbf{S}$ from 1. The elements of $\mathbf{S}^+$ and $\mathbf{S}^-$ are populated using the same approach for the elements of $\mathbf{M}^+$ and $\mathbf{M}^-$ and gamma distributions fit on $\mathbf{M}_p^+$, and $\mathbf{M}_p^-$.

The TPAC method is motivated in part by our previously developed Variance-adjusted Mahalanobis (VAM) method [34], which uses a modified Mahalanobis distance for cell-level gene set testing of single cell RNA-sequencing data. For the VAM approach, only positive distances, measured relative to the origin, are used and the sample covariance matrix is modified to capture just the technical component of gene expression variance. Similar to the use of gamma CDF values for the VAM method, the use of $F_{\gamma(\hat{\alpha}_k, \hat{\beta}_k)}()$ to generate TPAC scores has several important benefits: 1) it enables gene set inference for individual tumors, 2) it transforms the distances for gene sets of different sizes into a common scale, and 3) it produces scores that are bound between 0 and 1 and robust to large expression values.

## Choice of S, S⁺ or S⁻

The three different TPAC generated score matrices, $\mathbf{S}$, $\mathbf{S}^+$ and $\mathbf{S}^-$, capture distinct features of pathway dysregulation within each tumor and the choice of which scores to use will therefore vary depending on the analysis goals. For all matrices, large scores correspond to tumors that are more significantly dysregulated relative to the corresponding normal tissue and are therefore more likely on average to be associated with a poor prognosis or advanced cancer stage.

- $\mathbf{S}^+$: Large values in $\mathbf{S}^+$ correspond to tumors where expression of gene set members is elevated relative to the associated normal tissue. The use of normal tissue-specificity to adjust sample variances (as detailed in Table 2 above) will prioritize expression differences for genes that are normally surpressed in the associated normal tissue, i.e., a tumor is considered more dysregulated if genes that are expressed at a low level in the associated normal tissue relative to other tissues are up-regulated in the tumor.

- $\mathbf{S}^-$: Large values in $\mathbf{S}^-$ correspond to tumors where expression of gene set members is down-regulated relative to the associated normal tissue. In constrast to the impact on $\mathbf{S}^+$, the use of normal tissue-specificity to adjust sample variances leads to larger $\mathbf{S}^-$ values when tissue-specific genes are down-regulated in the tumor, i.e., a tumor is considered more dysregulated if genes that are expressed at a high level in the associated normal tissue relative to other tissues are down-regulated in the tumor.

- $\mathbf{S}$: Large values in $\mathbf{S}$ correspond to tumors where expression of gene set members exhibit a combination of up and down-regulation relative to the associated normal tissue.

For most of the analysis results presented below, we use the scores in the **S** matrix since these capture a wider range of dysregulation patterns.

## TPAC implementation

The TPAC method detailed above is implemented by the TPAC R package that is available on CRAN (https://cran.r-project.org/web/packages/TPAC). The requires normal tissue expression data from the HPA is made available via the TPACData R package, which is imported by TPAC and is also available on CRAN (https://cran.r-project.org/web/packages/TPACData). Readers interested in using the TPAC method should start by reviewing the liver cancer vignette (included in S2 Text and embedded in the TPAC package) which illustrates the computation of TPAC scores for TCGA liver cancer RNA-seq data using MSigDB Hallmark gene sets. The TPAC R package documentation contains complete details on the supported functions.

## Comparison methods

We compared the performance of the TPAC method against four existing single sample gene set testing techniques: GSVA [35], ssGSEA [36], the z-scoring method of Lee et al. [37] (referred to throughout the reminder of the paper as the 'z-scoring method'), and GRAPE [38]. We also included results for a 'null' version of TPAC as a negative control (i.e., TPAC scores with permuted sample labels to break any associations with cancer prognosis or stage), and a version of TPAC that does not use tissue-specific weights to adjust gene expression sample variances.

GSVA and ssGSEA are both widely used competitive techniques that generate sample-level scores using a Kolmogorov-Smirnov (KS) like random walk statistic computed on the gene ranks within each sample following some form of gene standardization across the samples. The z-scoring method is a simple self-contained technique that fits a standard normal distribution to the average of set genes across all samples. For GSVA, ssGSEA and the z-scoring method, the implementation in the GSVA R package was used with default parameter settings. Like the z-scoring technique and TPAC, the GRAPE method only uses the expression values for genes within the evaluated set. Similar to TPAC, and unique among the four comparison methods, GRAPE generates scores for each sample by comparison with a reference expression profile. In particular, GRAPE scores are based on the distance between a binarized version of set gene expression value for a given sample and a binarized reference template profile (or profile collection). For GRAPE, the binarization of a gene set with $m$ members results in a $m(m-1)/2$ length binary vector whose elements are indicator variables for an inequality operator applied to all gene pairs, i.e., the value for the $i, j$ comparison is 1 ($g_i < g_j$) with ties resolved randomly, and the distance between binary vectors are computed as the weighted average absolute difference of the elements. For the results in this paper, GRAPE scores were computed using the *makeGRAPE_psMat()* function in the GRAPE R package with the reference template set to the HPA expression profile for the associated normal tissue with all other parameters set to default values.

Although GRAPE is the most similar of the comparison methods to TPAC, there are a number of important differences between the two approaches: 1) the binarization approach used by GRAPE is equivalent to a rank-based representation whereas TPAC measures distances directly on the normalized expression values, 2) GRAPE computes distances as the averaged weighted absolute difference between binarized vectors and thereby ignores direction; TPAC separately models increased and decreased expression in the sample relative to the reference, 3) TPAC scores have a valid null distribution so can be used for inference; GRAPE

scores do not have a probabilistic interpretation, and 4) TPAC is specifically designed for the analysis of cancer gene expression data and adjusts scores based on gene normal tissue-specificity; although the GRAPE paper included TCGA analysis results, the method is not specifically optimized for the analysis of cancer data, the reported analyses did not score tumor expression data relative to the associated normal tissue and the method does not account for normal tissue-specificity.

## TCGA analyses

The TPAC method and the comparative methods outlined above in Section were used to generate single sample gene set scores for TCGA RNA-seq data from the 21 cohorts listed in Table 1 and the 50 gene sets from the MSigDB Hallmark collection. The TPAC **S** matrix scores and scores from the comparative methods were used for the following analyses:

- **Landscape of pan-cancer gene set dysregulation**: Single sample gene set scores for tumors from all TCGA cohorts and Hallmark gene sets were clustered and visualized. This analysis was performed to explore the pattern of gene set dysregulation across multiple cancer types.

- **Survival analysis**: Univariable Cox proportion hazards models were fit for each cohort using progression free interval (PFI) as the outcome and single sample gene set scores as a single predictor variable. The rationale for this analysis is an assumption that cancer prognosis is associated with the level of transcriptional dysregulation of important biological processes (as represented by Hallmark gene sets). Given ths assumption, we expected to find more statistically significant associations between TPAC scores and PFI than for the scores generated by comparative single sample gene set testing methods.

- **Tumor/lymph node stage analysis**: For each TCGA cohort, a Wilcoxon rank sum test was performed on the single sample scores generated by each method for all Hallmark gene sets. For the analysis of tumor stage, the TCGA tumor stage was discretized as T01 vs non-T01. For the analysis of lymph node stage, the TCGA lymph node stage was discretized as N0 vs non-N0. Similar to the rationale for performing a survival analysis using single sample gene set scores, this analysis was motivated by the assumption that transcriptional dysregulation is associated with cancer progression (as represented by tumor stage).

- **Single tumor inference**: To evaluate the use of TPAC scores for tumor-level inference, the CDF values in the **S** matrices for all cohorts were converted to p-values. False discovery rate (FDR) values were then computed for the family containing all gene set scores across the 21 TCGA cohorts and 50 Hallmark gene sets using the method of Benjamini and Hochberg method [39].

- **Kaplan-Meir analysis**: To explore the association between TPAC score significance and cancer prognosis, a Kaplan-Meir analysis was performed for the TCGA KIRP cohort and the progression-free interval (PFI) outcome with patients stratified according to the significance of TPAC score for the MSigDB Hallmark MYC Targets V1 gene set. Significance was determined according to whether the FDR value associated with the TPAC score was $< 0.25$ where the family of hypotheses included the TPAC scores for all 50 Hallmark gene sets for all 321 KIRP samples with PFI data (16,050 total hypotheses).

- **Transcription factor activity**: To explore the biological features represented by TPAC gene set dsyregulation scores, we estimated sample-level transcription factor (TF) activity from the TCGA gene expression data for all analyzed cohorts using the decoupleR method [40]. Following the logic in the 'Transcription factor activity inference in bulk RNA-seq' vignette

associated with the decoupleR Bioconductor R package, this analysis executed the decoupleR method using *run_wmean()* function with transcription factor target information obtained from the Collection of Transcriptional Regulatory Interactions (CollecTRI) database [41]. This decoupleR analysis generated a matrix of sample-level activity estimates for 618 TFs for all tumors in the analyzed TCGA cohorts. Spearman rank correlations were then computed between the TF activity estimates and the TPAC scores for all MSigDB Hallmark gene sets.

## Results

### Inference with TPAC scores maintains type I error control

To explore the statistical properties of the TPAC method, normalized count data was simulated under both the null and alternative hypotheses. To estimate the type I error rate, we generated a 10,0000-by-100 matrix of independent Poisson random variables with $\lambda = 5$ to which library size normalization was applied. For this null simulation, p-values derived from the **S** matrix generated by TPAC for a single set containing the first 20 variables maintained a type I error rate of 0.0527 at $\alpha = 0.05$. To estimate power, a similar 10,000-by-100 matrix of Poisson counts was simulated with a constant offset of $\lambda * \delta$ added to the values for the first 1,000 samples and 20 variables with $\delta \in \{0.1, 0.2, \ldots, 2\}$. Fig 1 illustrates the estimated empirical power for different effect sizes at $\alpha = 0.05$.

### TPAC scores reveal pan-cancer landscape of gene set dysregulation

Fig 2 illustrates the overall pattern of **S** matrix TPAC scores for the MSigDB Hallmark gene sets across tumors from all 21 evaluated TCGA cohorts (a similar heatmap for GSVA scores is included as Fig A in S1 Text). As seen in this figure, tumors cluster into four primary groups according to TPAC scores (these clusters are annotated along the bottom of the heatmap):

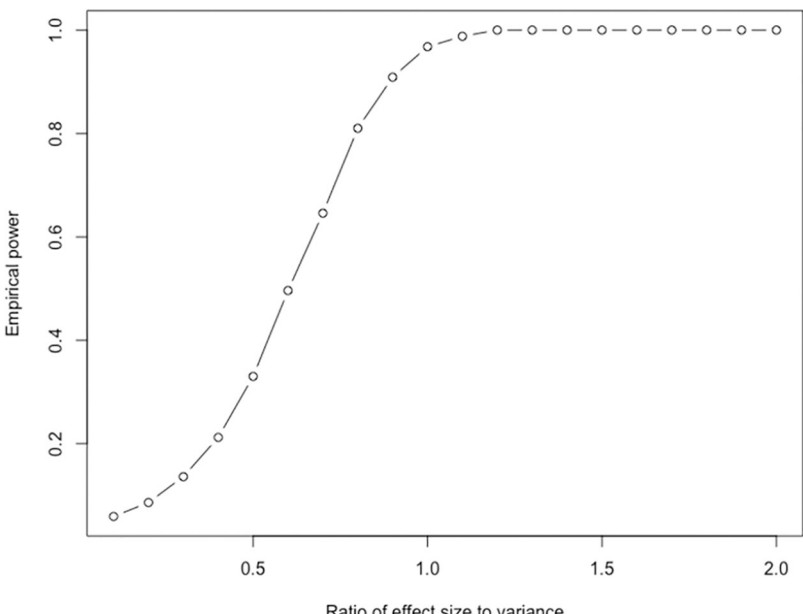

**Fig 1. TPAC empirical power for different simulated effect sizes.** The simulation model is detailed in section 'Inference with TPAC scores maintains type I error control'.

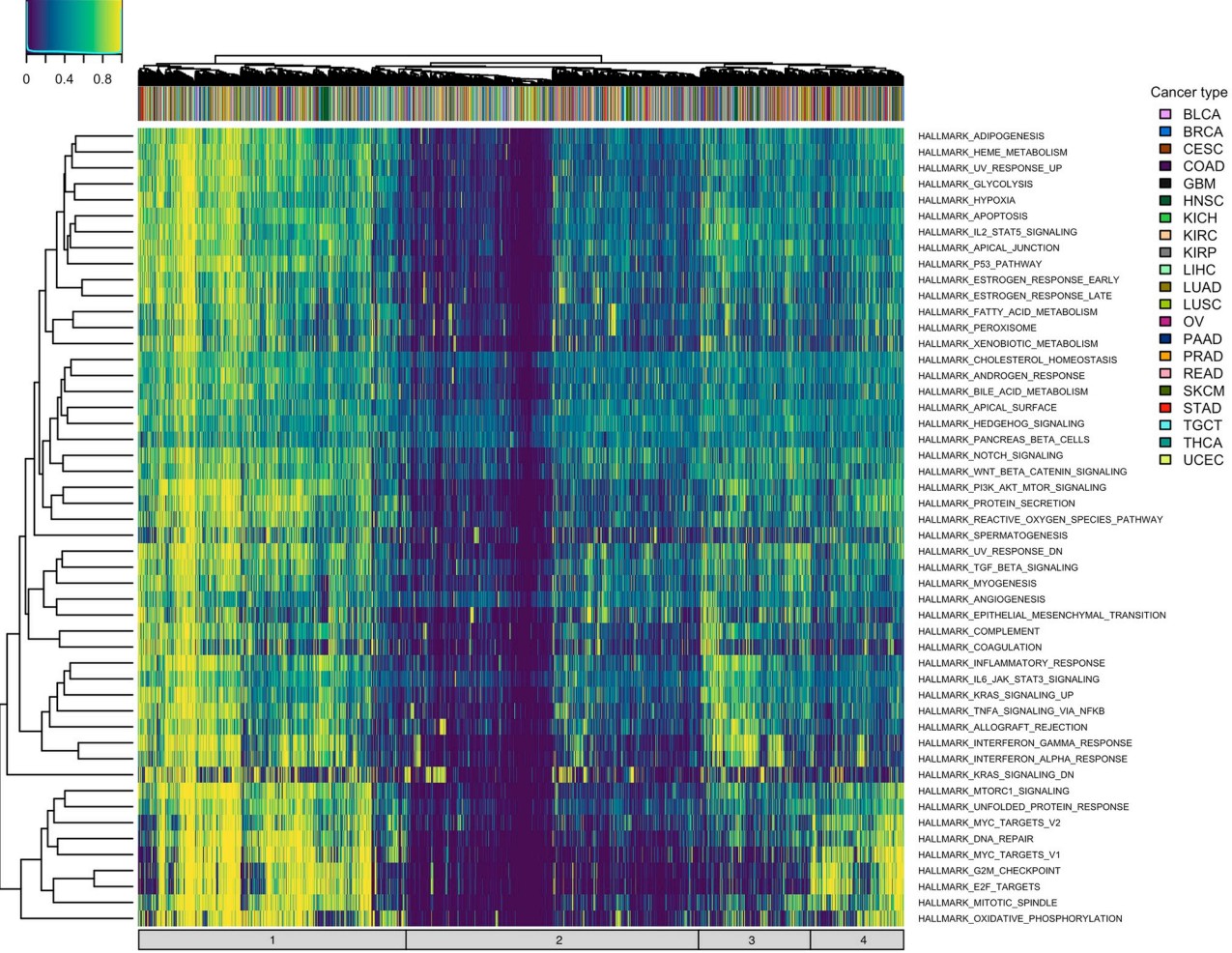

**Fig 2. Heatmap illustrating the pan-cancer distribution of S matrix TPAC scores for the MSigDB Hallmark gene sets.** Annotations along the top reflect cancer type and annotations on the bottom represent the four main types of dyregulation pattern (1: overall dysregulation across all gene sets, 2: minimal dysregulation, 3: immune signaling dysregulation, and 4: proliferation dysregulation).

1. **Overall dysregulation**: Tumors that are highly dysregulated across all Hallmark gene sets (i.e., tumors represented by the left-most columns in the heatmap)

2. **Minimal dysregulation**: Tumors that exhibit limited gene expression dysregulation (i.e., tumors represented by columns in the middle of the heatmap).

3. **Immune signaling dysregulation**: Tumors that show pronounced dysregulation among gene sets related to immune cell signaling (i.e., tumors represented by columns to the immediate right of the minimally dysregulated tumors)

4. **Proliferation dysregulation**: Tumors that show pronounced dysregulation among proliferation gene sets (i.e., tumors represented by the right-most columns in the heatmap)

## TPAC scores have more significant associations with cancer prognosis than scores generated by comparative methods

Fig 3 illustrates the distribution of p-values from univariable Cox proportional hazards models fit for each TCGA cohort that use progression-free interval (PFI) as the outcome. Separate Cox

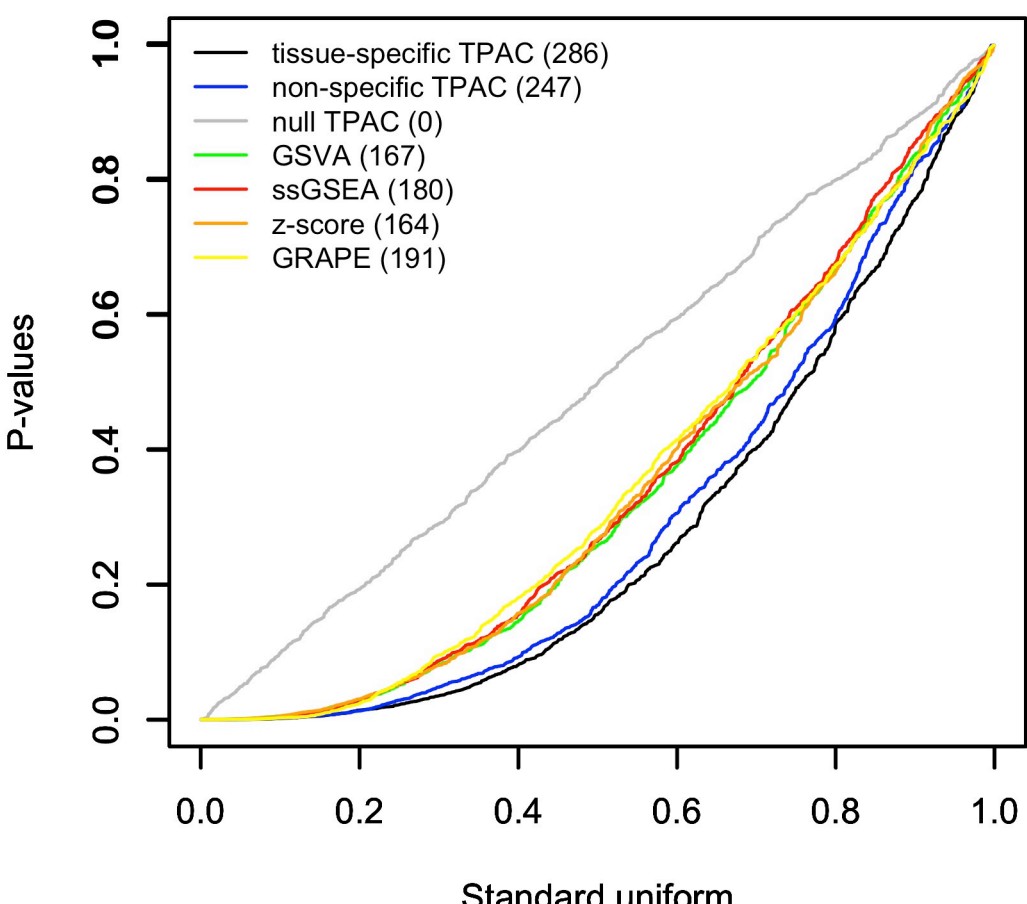

**Fig 3. Q-Q plot comparing p-values from Cox proportional hazards models that use single sample gene set scores generated by TPAC and each of the comparison techniques as the single predictor and PFI as the outcome against the $U(0, 1)$ distribution expected under the null.** For each evaluated method, a separate test is performed for all 50 MSigDB Hallmark pathways for each of the 21 analyzed TCGA cancer types for a total of 1,050 tests per method. The results for each evaluated single sample gene set testing method are plotted separately with the number of hypothesis tests out of a family of 1,050 tests associated with FDR values $\leq 0.1$ listed in paratheses after the method in the legend.

models were fit for each combination of TCGA cohort, Hallmark gene set, and gene set testing method with the distribution of all p-values associated with each analysis methods plotted as a separate curve. As shown in the figure, models using TPAC-generated gene set predictors provided the most significant associations (286), as determined by Cox models with FDR values $\leq 0.1$. As expected, the null version of TPAC had 0 significant results. The results for each TCGA cohort are visualized in separate Q-Q plots in Fig C in S1 Text, which demonstrates significant variability in prognositic signal and relative method performance across the different cancer types. The magnitude and direction of the PFI associations for the TPAC-generated gene set scores are visualized in Fig D in S1 Text. As illustrated by Fig D in S1 Text, gene set dysregulation is usually associated with an unfavorable cancer prognosis and the strength of the association varies across the TCGA cohorts.

## TPAC scores are strongly associated with tumor and lymph node stage

Fig 4 and Fig F in S1 Text provide a similar visualization as Fig 3 and Fig D in S1 Text of the p-values from statistical models based on single sample gene set scores computed using TPAC and

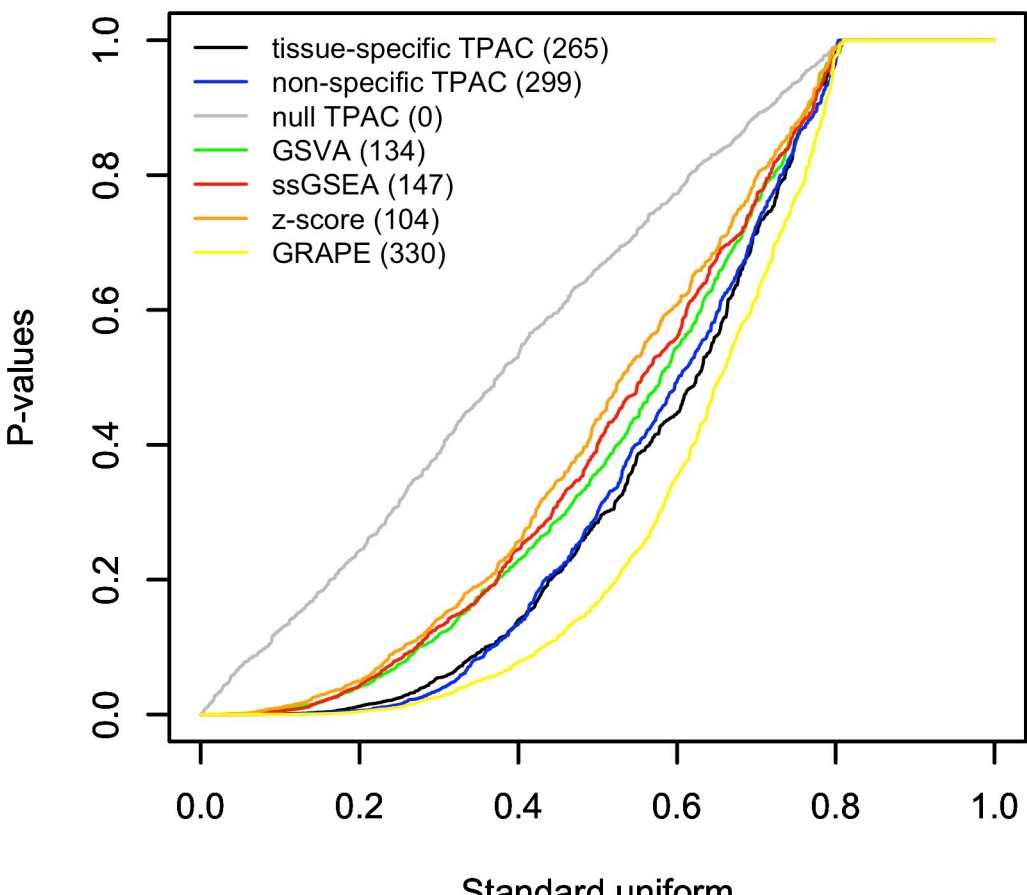

**Fig 4. Q-Q plot comparing the distribution of p-values from Wilcoxon rank sum tests comparing single sample gene set scores generated by TPAC and each of the comparison techniques for tumors with stage T01 vs. the scores for tumors with higher stages.** For each evaluated method, a separate test is performed for all 50 MSigDB Hallmark pathways for each of the 21 analyzed TCGA cancer types for a total of 1,050 tests per method. The results for each evaluated single sample gene set testing methods are plotted separately with the number of hypothesis tests out of a family of 1,050 tests associated with FDR values ≤ 0.1 listed in paratheses after the method in the legend.

the other comparison methods. For these figures, the p-values capture the association between gene set scores and discretized tumor stage (T01 vs. other) as evaluated using a Wilcoxon rank sum test. For these analyses, the scores generated by GRAPE generate the most significant results (330) with the TPAC variants a close second (299 and 265) and all other comparison methods producing a markedly smaller number of significant results (104, 137, and 147). Importantly, the two methods that generate scores based on the deviation in gene expression between each tumor and the corresponding normal tissue (i.e., TPAC and GRAPE) have substantially more significant associations relative to techniques that do not use a normal tissue reference. It is interesting to note that the version of TPAC that does not include an adjustment for tissue-specificity outperforms the version that includes the tissue-specific adjustment. The results for each TCGA cohort are visualized in separate Q-Q plots in Fig E in S1 Text.

Similar to Fig 4 and Fig F in S1 Text, Fig 5 and Fig H in S1 Text visualize the association between single sample gene set scores and lymph node stage associated with each tumor. Similar to the tumor stage analysis, the GRAPE method generated the most significant results (216) for the lymph node stage analysis with TPAC second (173). In this case, the standard TPAC

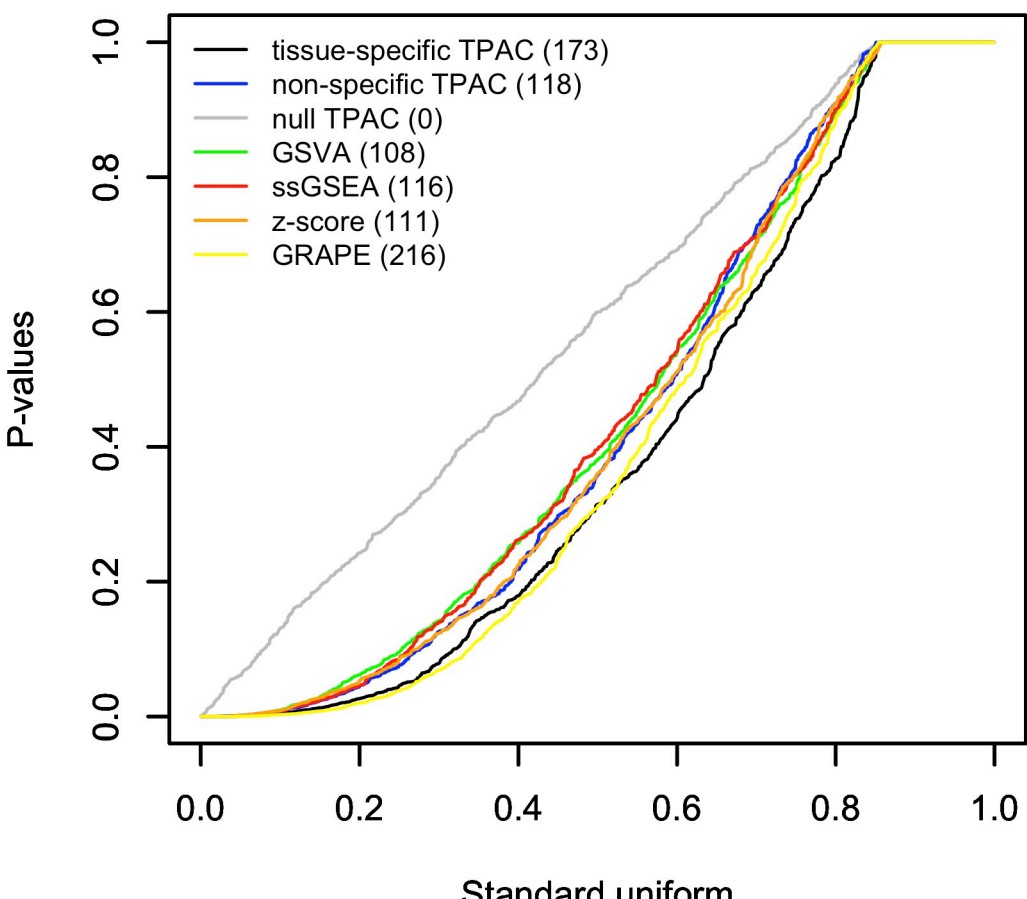

**Fig 5. Q-Q plot comparing the distribution of p-values from Wilcoxon rank sum tests comparing single sample gene set scores generated by TPAC and each of the comparison techniques for tumors associated with lymph node stage N0 vs. the scores for tumors associated with higher lymph node stages.** For each evaluated method, a separate test is performed for all 50 MSigDB Hallmark pathways for each of the 21 analyzed TCGA cancer types for a total of 1,050 tests per method. The results for each evaluated single sample gene set testing methods are plotted separately with the number of hypothesis tests out of a family of 1,050 tests associated with FDR values ≤ 0.1 listed in paratheses after the method in the legend.

generated substantially more significant associations (173) than the non-tissue specific TPAC (118). The results for each TCGA cohort are visualized in separate Q-Q plots in Fig G in S1 Text.

## Sample-level inference on TPAC scores highlights the significant elements of pan-cancer gene set dysregulation

Fig 6 illustrates the use of TPAC scores for tumor-specific inference regarding gene set dysregulation. To generate this heatmap, the CDF values in the **S** matrices for all TCGA cohorts were converted to p-values. False discovery rate (FDR) values were then computed for the family containing all gene set scores across the 21 TCGA cohorts and 50 Hallmark gene sets (461,900 total hypotheses) using the method of Benjamini and Hochberg method [39]. For cells corresponding to FDR values ≥ 0.3, the TPAC scores are set to 0 and the modified TPAC scores are then visualized as a heatmap. For this analysis, 2.6% of the TPAC scores (11,797 individual scores) were significant at an FDR threshold of 0.3.

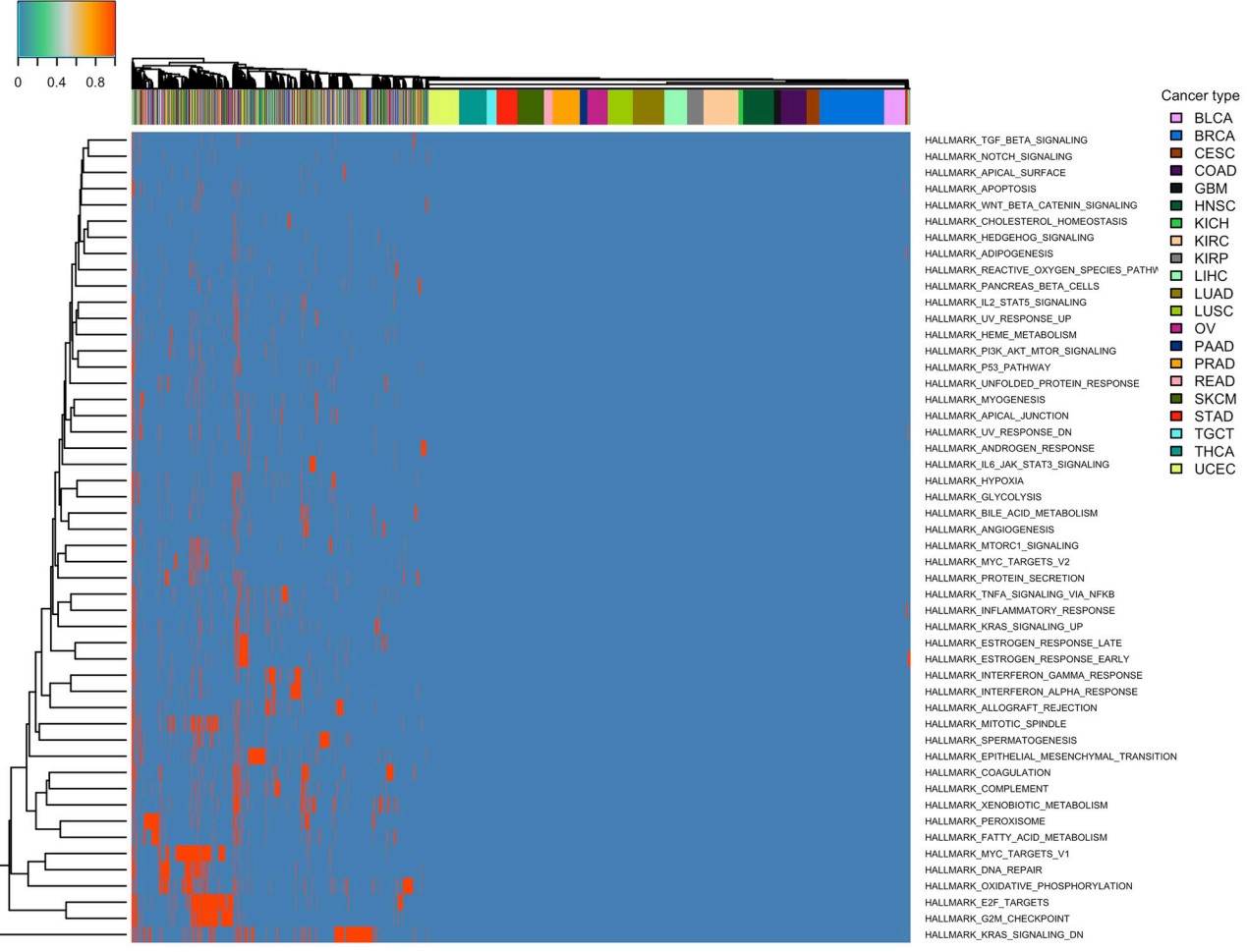

**Fig 6. Illustration of the pan-cancer significance of S matrix TPAC scores for the MSigDB Hallmark gene sets.** This figure visualizes the same data as Fig 2 but with each TPAC scores whose associated FDR value is $\geq$ 0.3 set to 0 (X total hypotheses).

## TPAC score statistical significance effectively stratifies patients according to cancer prognosis

Fig 7 visualizes the use of tumor-level inference for survival analysis. Specifically, the TPAC scores for the TCGA KIRP cohort and Hallmark MYC Targets V1 gene set were discretized according to an FDR threshold of 0.25 (16,050 total hypotheses) and these discretized values were then used for a Kaplan-Meir analysis relative to patient PFI. As shown in this plot, tumors with significant dysregulation of the MYC Targets V1 gene set have a significantly worse prognosis than tumors lacking significant dysregulation. Because only the TPAC method supported score-level inference, comparable results cannot be generated for the other evaluated methods.

## Association between TPAC scores and transcription factor activity illuminates the regulatory impact of tumorigenesis

Fig 8 visualizes the rank correlation between tumor-level transcription factor (TF) activity values, as estimated using the decoupleR [40] method, and overall TPAC scores for each analyzed

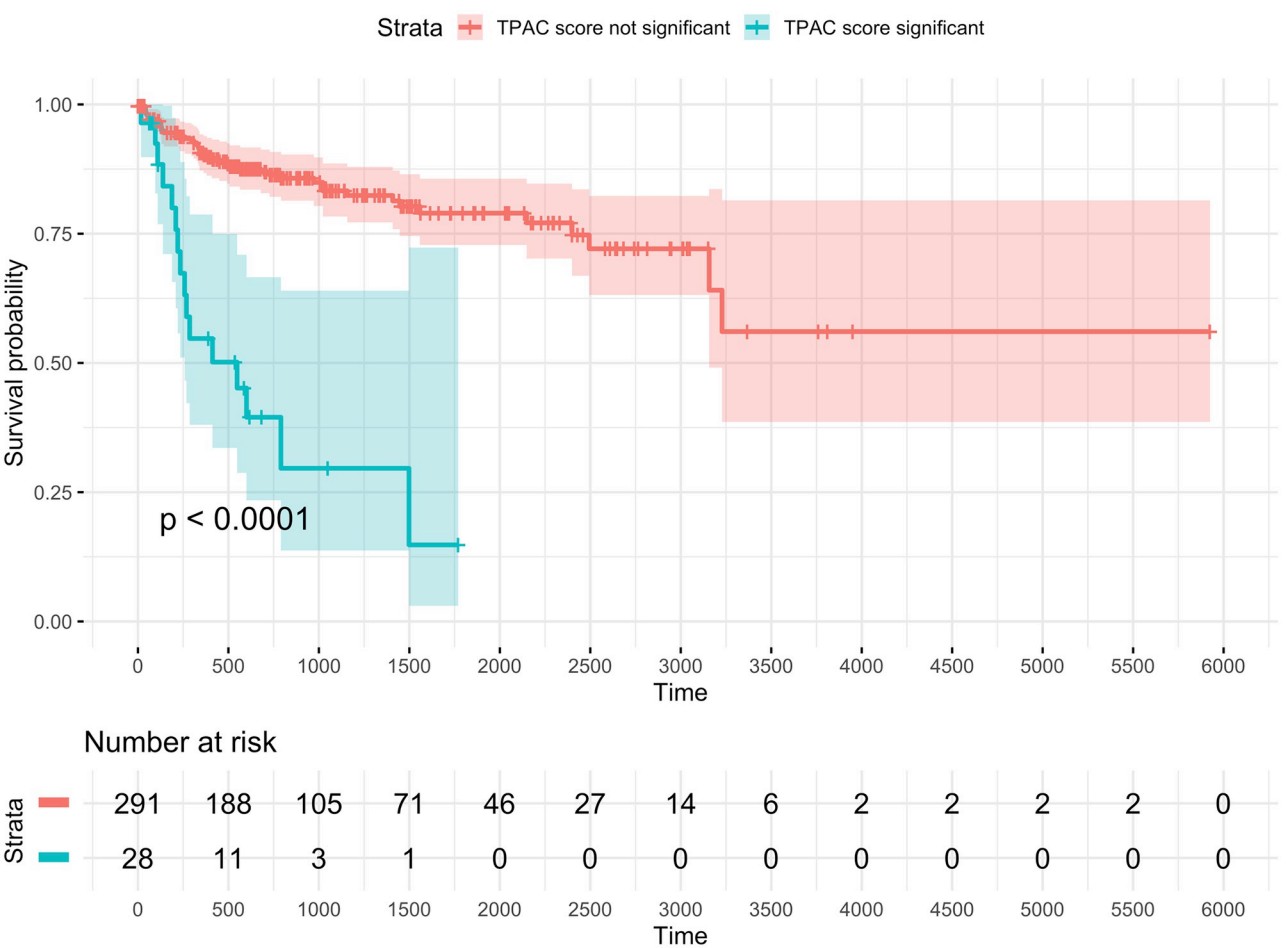

**Fig 7. Kaplan-Meir plot for TCGA KIRP cohort and progression-free interval (PFI) outcome with patients stratified according to the significance of the TPAC score for the MSigDB Hallmark MYC Targets V1 gene set.** Significance was determined according to whether the FDR value associated with the TPAC score was < 0.25 where the family of hypotheses included the TPAC scores for all 50 Hallmark gene sets for all 321 KIRP samples with PFI data (16,050 total hypotheses).

TCGA cohort. Each cell represents the average rank correlation across all 50 MSigDB Hallmark gene sets; only the 25 TFs with the largest average absolute correlation are shown. These results reveal that the estimated activity for most TFs do not have a consistent association with TPAC scores across all Hallmark gene sets and that the pattern of association varies significantly between the different cancer types. However, the cases where TF and TPAC scores do have a consistent direction of association for a cancer type across all of the Hallmark gene sets, e.g, KLF13, THAP11 and IRF7 for GBM, are consistent with prior findings on TFs whose activity is linked to cancer (KLF13 [42], THAP11 [43], and IRF7 [44]). Fig F in S1 Text shows the per-gene set correlation averaged across all cohorts. In this case, the rank correlations are quite small when averaged across all cohorts but, for the top TFs, show a consistent direction of association across all Hallmark gene sets. The TFs captured in Fig F in S1 Text also have known associations with cancer (e.g., KLF4 [45] and PAX2 [46]).

As expected, larger magnitude TF/TPAC correlations are observed when not averaging over cancer types or gene sets. Fig G in S1 Text and Fig H in S1 Text visualize these correlations for the KIRP and KIRC cohorts. Looking specifically at the Hallmark MYC Targets V1

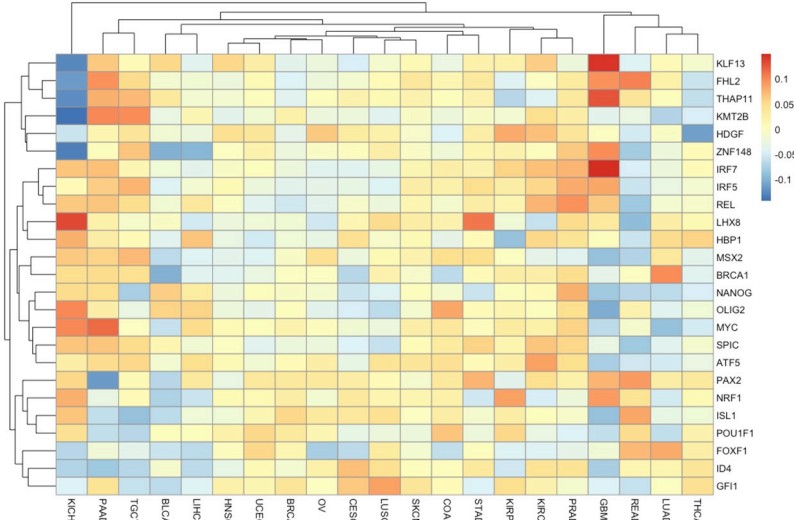

**Fig 8. Association between transcription factor (TF) activity, as estimated using the decoupleR method, and TPAC scores.** Each cell represents the average rank correlation between overall TPAC scores and TF activity estimates for one of the TCGA cohorts where averaging is performed across all 50 MSigDB Hallmark gene sets. Results are only shown for the 25 TFs with the largest average absolute correlation.

gene set that was used to generate the Kaplan-Meir plot in Fig 7, the 10 TFs with the largest positive and negative rank correlations in the KIRP cohort are listed in Table B in S1 Text and provide some insight into potential regulatory mechanisms underlying the significant association with TPAC score significance for this gene set and KIRP cancer prognosis. Similar to the results shown in Fig 8, Fig F in S1 Text and Fig G in S1 Text, these TFs have known links to cancer (e.g., SIX4 [47] and PURA [48]).

## Discussion

### Novel features of the TPAC method

The TPAC method incorporates several novel features not found in existing single sample gene set testing methods:

- TPAC scores capture the deviation in gene set expression between each analyzed tumor and the associated normal tissue using a novel modified Mahalanobis distance metric. While the GRAPE method also generates scores based on the distance between the expression profile of each sample and a reference, it is not specific to cancer, and, prior to the evaluation in this paper, had not been used to score tumor expression relative to normal tissue.

- TPAC adjusts for gene normal tissue-specificity using the weighting scheme detailed in Table 2, which improves the association between the generated gene set scores and both cancer prognosis and lymph node stage.

- The TPAC method generates separate scores that capture the positive ($S^+$), negative ($S^-$) and overall ($S$) deviations in tumor expression relative to normal tissue. These direction-based scores not only support the use of tissue-specific weights (as detailed in Table 2) but also enable a range of downstream analyses (as detailed in the 'Choice of $S$, $S^+$ or $S^-$' section). By contrast, the GRAPE method generates scores based on the average weighted absolute

difference between binarized vectors, which does not distinguish between positive and negative deviations.

- The modified Mahalanobis distances used by TPAC have a null distribution that approximately follows a gamma distribution, which enables both population and sample-level inference. While this probabilistic score model is shared by our previously developed VAM technique [34], the VAM method is not appropriate for the TCGA analyses explored in this paper since it focuses on single cell RNA-seq data and does not support the comparison against a reference profile.

## Benefits of TPAC for the analysis of cancer transcriptomics

The novel features of the TPAC method underpin several important analytical benefits:

- TPAC scores capture tissue-independent features of the tumor transcriptome. A differential expression analysis of two cancer types will generate results that largely mirror the output from a comparison of the associated normal tissues (this effect is visualized in Fig 4 of our prior paper [21]). This strong tissue type signal makes it challenging to disentangle the contribution of normal tissue biology from the impact of mutagenic processes when analyzing cancer transcriptomic data. By quantifying the deviation of each tumor from the associated normal tissue, the TPAC method help identify the tissue-independent transcriptomic impact of a given cancer. This property of TPAC scores is demonstrated in Fig 2 by the fact that tumors cluster according to common dysregulation patterns rather than by cancer/tissue type.

- TPAC scores have the strongest association with cancer prognosis among comparable techniques. Under the assumption that cancer aggressiveness is positively correlated with the disruption of important biological processes (as represented by the MSigDB Hallmark gene sets), we expected to find statistically signficant associations between TPAC scores for many of the Hallmark gene sets and cancer prognosis as represented by the progression free interval (PFI). As shown in Fig 3, TPAC scores have the most significant associations with PFI (286 vs. a range of 164–191 for the other techniques). Importantly, the standard version of TPAC that adjusts for normal tissue-specificity has more significant associations (286) than a variant of TPAC that does not include the tissue-specificity adjustment (247). The strong association between TPAC scores and cancer prognosis is also illusrated by Fig 7, where stratification of KIRP patients according to the statistical significance of the TPAC score for Hallmark MYC Targets V1 gene set reveals a dramatic difference in PFI (i.e., significant dysregulation of this gene set relative to normal kidney tissue is associated with a much shorter PFI).

- TPAC scores have a strong association with both tumor and lymph node stage. Similar to our assumption regarding a link between cancer aggressiveness and pathway dysregulation, we also assumed that cancer progression is marked by an increased dysregulation in key cellular pathways. Given this assumption, we expected to find a statistically significant association between TPAC scores and both tumor and lymph node stage. In constrast to the PFI analysis, scores from the GRAPE method had the most significant associations with tumor stage (330 vs. a range of 104–299 for the other methods) and lymph node stage (216 vs. a range 108–173). Although TPAC did not produce the most significant associations, its performance was close to GRAPE (299 vs. 330 for tumor stage; 173 vs. 216 for lymph node stage) and markedly above the other techniques (299 vs. a range of 104–147 for tumor stage; 173 vs. a range of 108–116 for lymph node stage). Importantly, the best performing methods

(GRAPE and TPAC) both generate scores based on the deviation between tumor expression and expression in the associated normal tissue. In this context, it is worth noting that the version of GRAPE used in the comparative evaluation (i.e., GRAPE with HPA normal tissue expression as the reference profile) is distinct from the version detailed in the GRAPE paper. This use of associated normal tissue data was not explored in the original GRAPE paper and can be viewed as part of the contribution of this manuscript.

- TPAC scores are associated with the activity of cancer-linked transcription factors (TF). As illustrated in Fig 8 and Fig I in S1 Text, Fig J in S1 Text and Fig K in S1 Text, TPAC scores are associated with the estimated activity of TFs that have been experimentally shown to be up or down-regulated within human cancers. Although the strength of these associations is low and there is significant heterogeneity across different tumor types, the results demonstrate that TPAC scores may be helpful in characterizing the regulatory program driving biological process dysregulation during tumorigensis.

## TPAC limitations

When interpreting the results in this manuscript or considering adoption of TPAC method, readers should keep several important limitations in mind. First, the results presented in this paper analyzed TCGA bulk RNA-seq data for solid human cancers using expression data for the associated normal tissues from the HPA. While a common processing pipeline was used on both the TCGA and HPA data, batch effects may still exist. Use of the TPAC method (as implemented in the TPAC R package) uses HPA normal tissue bulk RNA-seq data so users must ensure that a equivalent normalization process is employed if analyzing non-TCGA RNA-seq data. A related limitation is that the TPAC R package only supports the analysis of bulk RNA-seq data for 21 solid human cancer types associated with the 18 normal tissue types whose HPA data is embedded in the package; analysis of other cancer types requires the custom integration of expression data for the relevant normal tissue from HPA or another repository. Analysis of cancer single cell RNA-seq data is not currently supported and will require the development of a cell type, vs. tissue type, reference model.

A second important limitation is that the relative performance of the TPAC method varies considerably across the various analyzed cancer types. As a consequence, potential users of the TPAC method should not assume that the overall analysis results presented in Figs 3, 4 and 5 will necessarily hold for a more focused analysis. The cancer type-specific results in Fig C in S1 Text, Fig E in S1 Text and Fig G in S1 Text should be consulted to understand the expected performance on a specific cancer type and outcome measure. When interpreting the TPAC performance reported in this paper, readers should also keep in mind that TPAC only supports unordered gene sets (i.e., the relationships between genes within a pathway are not considered), that the normal tissue-specific weighting used by TPAC may not improve performance in all cases (e.g., the association between TPAC scores and tumor stage shown in Fig 4), and that an improvement in the number of significant associations (as shown for PFI in Fig 3 is only beneficial as long as the type I error rate is maintained.

## Conclusion

Cancer biology is highly tissue-specific: most cancer-driving somatic alterations occur in just a limited number of tissues and inherited mutations frequently have a tissue-specific functional impact. As we have explored in prior work [21], cancer tissue-specificity can be leveraged to improve the power and accuracy of cancer genomic analyses. To leverage the associations between gene activity in normal and malignant tissue for gene set testing, we have developed a

new single sample gene set testing method for tumor-derived transcriptomics data named TPAC (tissue-adjusted pathway analysis of cancer). The TPAC method uses the normal tissue-specificity of human genes to compute a robust multivariate distance score that quantifies gene set dysregulation in each profiled tumor and can be used for both population and sample-level inference. As demonstrated through an analysis of TCGA RNA-seq data, TPAC gene set scores are more strongly associated with patient prognosis than the scores generated by existing single sample gene set testing methods.

## Supporting information

**S1 Text. Includes supplemental methods and results.**
(PDF)

**S2 Text. Contains a vignette illustrating the analysis of TCGA liver cancer RNA-seq data using MSigDB Hallmark gene sets.**
(PDF)

## Acknowledgments

We would like to thank HPA group for providing access to the HPA.normal.FPKM.GDCpipe-line.csv file.

## Author Contributions

**Conceptualization:** H. Robert Frost.

**Data curation:** H. Robert Frost.

**Formal analysis:** H. Robert Frost.

**Funding acquisition:** H. Robert Frost.

**Investigation:** H. Robert Frost.

**Methodology:** H. Robert Frost.

**Project administration:** H. Robert Frost.

**Resources:** H. Robert Frost.

**Software:** H. Robert Frost.

**Supervision:** H. Robert Frost.

**Validation:** H. Robert Frost.

**Visualization:** H. Robert Frost.

**Writing – original draft:** H. Robert Frost.

**Writing – review & editing:** H. Robert Frost.

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
