## [Decision Letter · Decision Letter 0]

24 Jul 2023

Dear Dr. Frost,

Thank you very much for submitting your manuscript "Tissue-adjusted pathway analysis of cancer (TPAC)" for consideration at PLOS Computational Biology.

As with all papers reviewed by the journal, your manuscript was reviewed by members of the editorial board and by several independent reviewers. In light of the reviews (below this email), we would like to invite the resubmission of a significantly-revised version that takes into account the reviewers' comments.

We cannot make any decision about publication until we have seen the revised manuscript and your response to the reviewers' comments. Your revised manuscript is also likely to be sent to reviewers for further evaluation.

Sincerely,

Martin H. Schaefer

Guest Editor

PLOS Computational Biology

Ilya Ioshikhes

Section Editor

PLOS Computational Biology

Reviewer's Responses to Questions

**Comments to the Authors:**

Reviewer #1: Summary: Robert Frost presents a new single sample pathway analysis method for the analysis of gene expression data from 21 TCGA cancer types. The method quantifies pathway dysregulation in individual TCGA tumors when compared to the pathway activity in normal tissue. The method is implemented in R and made available via an R package on the author’s homepage. The data shows that the method controls the type I error rate in simulated data, and when compared to existing single-sample enrichment methods, TPAC produces stronger associations with clinical correlates. Strengths include method simplicity and apparent effectiveness in detecting associations between pathway activity and tumor stage and patient prognosis. Weaknesses include that the method seems to be specifically designed for TCGA data and it is not clear how the method would generalize to cancer gene expression data from other sources. Also, the organization of the manuscript needs to be improved.

Comments:

Manuscript organization:

- It is unusual to refer to display items in the Introduction. Table 1 and the corresponding paragraph referring to Table 1 should be moved to the beginning of the Results section

- In agreement with the journal guidelines, Results and Discussion should be divided into two sections. Currently there is no real discussion, the section is reporting merely on results, with no contextualization or interpretation guidance for the reader. The Discussion section is a very important section for most readers, providing an overview, linking the different parts of the manuscript, and helps to embed the findings in the existing literature.

- There are currently too many Figures (10). Consider creating composite figures (ie. thematically grouping figures 3 and 4 as subfigures A and B of one figure; and analogously for eg Figure 5 and 6). Or consider moving 3-5 figures to the Supplement.

- Availability + Implementation is currently pointed out at 5 different places in the manuscript (abstract, intro, method, conclusion, data availability). Although it is helpful to rather mention availability too often than too less, the typical place to mention availability + implementation would be Methods and in a dedicated availability paragraph at the end of the manuscript.

Comments by section:

- Title: The title is currently a bit bare bone. Maybe a more accessible title could be: “Tissue-adjusted pathway analysis of cancer (TPAC) increases association strength with tumor stage and patient prognosis” or something along these lines that would allow a reader to immediately grasp the benefits of the method in a cancer expression data analysis setting.

- Intro:

o The method is currently branded as a pathway analysis method, but it seems to be generally applicable to gene set testing (ie also to gene sets that are not pathways). Also: pathway analysis implies for many readers a certain extent of incorporation of regulatory relationships between the genes in a pathway (= topology- or network-based methods) which is not the case here. This distinction should be made clearer in the first paragraph of the introduction and throughout the manuscript.

o Claim: Ref 19 shows that “the ratio of gene expression in a specific tissue or cancer relative to the mean across multiple tissues or cancers, can be used to improve survival analysis, the comparative analysis of distinct cancer types, and the analysis of cancer/normal tissue pairs” – how precisely does it improve these three analysis types?

o “Specifically, we found that genes enriched in normal tissues are more likely to be down-regulated in the associated cancer with elevated expression associated with a favorable cancer prognosis” -> This sentence is difficult to parse. Consider splitting into two sentences: 1) Genes that are highly-expressed in normal tissue are typically down-regulated in cancer; 2) elevated expression of these genes (in cancer?) is typically associated with improved prognosis. “Genes enriched” -> “Highly-expressed genes” or “Genes with increased expression”.

o Table 1: see comments for the Results section

o “has an accurate gamma approximation” -> follows approximately a gamma distribution

o “characterize the performance of TPAC” -> evaluate/assess the performance of TPAC

o Consider moving pointers to supplement and R package to the corresponding places in Methods and Results

- Methods:

o Why using normal tissue expression data from HPA as opposed to expression in adjacent normal tissues available from TCGA? How are batch effects taken into account when substracting normal tissue expression assayed by HPA from cancer tissue expression assayed by TCGA? These two might reside on somewhat different scales due to batch effects.

- Results:

o Table 1:

# “TCGA abbrev.” -> TCGA disease code

# It might make sense to restrict the computation of the Spearman rank correlation rho(cancer/norm) to something like the 500-1000 top genes, as computing correlations across something like 20k genes has the potential to dilute signals that are apparent on relevant subsets of the full gene vector.

# Gene weights are computed as the log fold-change of the mean expression in the normal tissue to the mean in all normal tissues. How many samples are there for each normal tissue? Or is there only one sample per tissue? If there were substantially different numbers of samples between normal tissues, taking the mean of all normal tissues could introduce a compositionality effect, where the normal tissues with many more samples would drive the resulting weights. That is similar to differential expression testing between clusters in single-cell RNA-seq data that is used for the identification of marker genes for each cluster.

o Figure 1:

# Type I error control and power analysis is carried out on simulated multivariate normal data, but there is no justification or reference to the literature that this is a good model for real data, in particular FPKMs as available from TCGA that TPAC is applied to in the manuscript

o Figure 2:

# It might make sense to label the gene set clusters in the heatmap as the main text refers to some of them in a not necessarily obvious way (“pathways related to immune signaling”, “dysregulation among proliferation pathwats”)

# It might make sense to spread out the heatmap over the full page width to be able to see whether there are any patterns in the tumor type clustering. From what I can see in the dense display, it seems there is little grouping by tumor / tissue type, which would argue against tissue-specific pathway-dysregulation? This could be discussed.

o Figure 3, 5, and 7:

# Whether or not one accounts for tissue-specificity in TPAC seems to hardly make a difference. Why is that? Discussion point.

# TPAC seems to yield more significant results than existing methods in terms of providing lower p-values? Why is that? Is that always beneficial or might this point at times to the method being too liberal in certain scenarios? Discussion point.

# The number of tests with FDR < 0.1 are shown in brackets. It would be helpful to also have the total number of tests eg in the figure caption.

o Figure 4, 6, and 8:

# Would it make sense to cluster the heatmaps to groups of tumor types and groups of hallmark sets that display similar behavior?

# Typo Figure 4: “rations” -> ratios

Software:

- Availability: TPAC is implemented as an R package that is available for download from the author’s homepage. I would strongly encourage making the package available through Github to enable routes for direct installation from github, as well as collaborative elements such as open issue tracking and pull requests. On the other hand, there are existing repositories such as CRAN for R packages in general and Bioconductor for bioinformatic R packages in particular that provide for adherence to R package standards, stable availability, and continuous integration with the R/Bioconductor package ecosystem. I would strongly encourage a submission of this package to Bioconductor.

- Package DESCRIPTION file: Doesn’t need to “depend” on MASS, seems sufficient to “import” from MASS

- I could install and use the package as described. However, I encountered the following error in example code of ‘tpacForCancer’ and ‘tpacForCollection’:

> library(TPAC)

# example taken from ?tpacForCancer

> # Simulate Gaussian expression data for 10 genes and 10 samples

> # (Note: cancer expression should be FPKM+1 for real applications)

> cancer.gene.expr=matrix(rnorm(100), nrow=10)

> # Create arbitrary Ensembl IDs

> gene.ids = c("ENSG00000000003","ENSG00000000005","ENSG00000000419",

+ "ENSG00000000457","ENSG00000000460","ENSG00000000938",

+ "ENSG00000000971","ENSG00000001036","ENSG00000001084",

+ "ENSG00000001167")

> colnames(cancer.gene.expr) = gene.ids

> # Define a collection with two disjoint sets that span the 10 genes

> collection=list(set1=gene.ids[1:5], set2=gene.ids[6:10])

> # Execute TPAC on both sets

> tpacForCancer(cancer.gene.expr=cancer.gene.expr, cancer.type="glioma",

+ gene.set.collection=collection)

Computing TPAC distances for 2 gene sets, 10 samples and 10 genes.

Min set size: 5, median size: 5

Error in stats::optim(x = c(0, 0, 0, 0, 1.01289132327529, 0, 0, 0, 0, :

L-BFGS-B needs finite values of 'fn'

Error in stats::optim(x = c(0, 0, 0, 0, 0, 0, 0, 0, 0, 0), lower = 0.01, :

non-finite value supplied by optim

In addition: Warning message:

In computeCDFValues(mahalanobis.sq = mahalanobis.sq$pos.dist.sq, :

Estimation of gamma distribution failed, defaulting p-values to 1

$S.pos

set1 set2

[1,] 0 0

[2,] 0 0

[3,] 0 0

[4,] 0 0

[5,] 0 0

[6,] 0 0

[7,] 0 0

[8,] 0 0

[9,] 0 0

[10,] 0 0

$S.neg

set1 set2

[1,] 0.31224115 0.2001183980

[2,] 0.96712990 0.7700834893

[3,] 0.29464712 0.5605427734

[4,] 0.45422304 0.4228654559

[5,] 0.01679889 0.6597519151

[6,] 0.21180767 0.6580781817

[7,] 0.71689591 0.7763170399

[8,] 0.82776827 0.0008710938

[9,] 0.29711699 0.6179639211

[10,] 0.88427436 0.9014183947

$S

set1 set2

[1,] 0.31218879 0.2001183980

[2,] 0.96691486 0.7700834893

[3,] 0.29461677 0.5605427734

[4,] 0.45398767 0.4228654559

[5,] 0.01687411 0.6597519151

[6,] 0.21187196 0.6580781817

[7,] 0.71641718 0.7763170399

[8,] 0.82921331 0.0008710938

[9,] 0.29708357 0.6179639211

[10,] 0.88385345 0.9014183947

Warning message:

In computeCDFValues(mahalanobis.sq = mahalanobis.sq$pos.dist.sq, :

- Estimation of gamma distribution failed, defaulting p-values to 1

- > tpacForCollection(gene.expr=gene.expr, mean.expr=mean.expr,

- + tissue.specificity=tissue.specificity, gene.set.collection=collection)

- Computing TPAC distances for 2 gene sets, 10 samples and 10 genes.

- Min set size: 5, median size: 5

- Error in stats::optim(x = c(7.18346325794129, 0, 2.88202834725928, 4.14166067879411, :

- L-BFGS-B needs finite values of 'fn'

- $S.pos

- set1 set2

- [1,] 0 0.13552825

- [2,] 0 0.53584159

- [3,] 0 0.35492329

- [4,] 0 0.01321637

- [5,] 0 0.89847009

- [6,] 0 0.08002146

- [7,] 0 0.89975993

- [8,] 0 0.51319159

- [9,] 0 0.91465787

- [10,] 0 0.57437133

-

- $S.neg

- set1 set2

- [1,] 0.4916428 0.99650402

- [2,] 0.8850201 0.43333510

- [3,] 0.7239927 0.03508627

- [4,] 0.4963750 0.38555402

- [5,] 0.7035225 0.00000000

- [6,] 0.6964786 0.46404505

- [7,] 0.7332576 0.59701028

- [8,] 0.1322641 0.27430659

- [9,] 0.4734590 0.00000000

- [10,] 0.5151094 0.77873917

-

- $S

- set1 set2

- [1,] 0.991709715 0.99600908

- [2,] 0.742783772 0.33296984

- [3,] 0.895400724 0.03504460

- [4,] 0.418454624 0.04620199

- [5,] 0.379853032 0.56313713

- [6,] 0.286837020 0.10994364

- [7,] 0.431133251 0.83025441

- [8,] 0.008457584 0.20672037

- [9,] 0.382973648 0.60957230

- [10,] 0.096726290 0.71563941

-

- Warning message:

- In computeCDFValues(mahalanobis.sq = mahalanobis.sq$pos.dist.sq, :

- Estimation of gamma distribution failed, defaulting p-values to 1

Reviewer #2: The paper “Tissue-adjusted pathway analysis of cancer (TPAC)” describes a pathway-analysis method. It uses the normal tissue-specificity of human genes to compute a robust multivariate distance score that quantifies pathway dysregulation in profiled tumors. The author addresses an important issue of highlighting pathways to study dysfunctionalities in human tissues, specifically in cancer.

Major comments:

1. The literature is not up to date. There are a few cited papers that were published after 2020, the latest paper cited was published in 2021. It is recommended to further explore and mention more recent studies in the field. For example, the author claimed that “current approaches leverage just tumor-specific genomic data and do not take into account gene activity in the associated normal tissues”. See for example, "Tissue-specific pathway association analysis using genome-wide association study summaries", Bioinformatics 2017.

2. The TPAC method is compared to GSVA, ssGSEA [28] Lee et al., [29], published in 2013, 2009, 2008. Comparison should be made to more recent methods, such as “GRAPE: a pathway template method to characterize tissue-specific functionality from gene expression profiles”, BMC Bioinformatics 2017.

3. The figures are unclear and should be reorganized in a way that allows readers to compare different cases. It is also recommended to elaborate more on the legend about the plots axes, the colors etc. None of the figures can stand on their own and could not be understood based on the legend. Also, if possible, consider uniting several figures into one with several panels. Fig. 2 is referred to as showing four clusters, but they are unclear.

4. The author claims that the TPAC method performed better than the 3 other methods. What were the comparison criteria? Has TPAC preformed significantly better? How was it measured? What were the differences?

5. It is not clear how the analyses that are described in the results section lead to the conclusions. More efforts should be made to clearly explain the rationale of each of the analyses and the conclusions that were made. It is also recommended that the titles for each section in the results will indicate on the conclusion of the analysis, rather than description (which can be used for figure names, for example).

6. The author used rich and diverse analyses. However, the methods should be written in a way that could be reproducible. It might be helpful to summarize the different parameters in a table with their meaning so readers can follow more easily.

7. It will be helpful to provide documentation and a description of the functions and methods available on the R package.

Minor

The pathways data used in the study are taken from MSigDB, which holds gene sets collections. It is important to clarify it as gene set rather than pathways, as the gene sets do not include information on protein-protein interactions or the direction of the pathway.

Reviewer #3: The author describes a single sample pathway analysis method for cancer

transcriptomics data, named tissue-adjusted pathway analysis of cancer (TPAC).

This method aims at leveraging information from normal tissue-specificity of human

genes to quantify pathway dysregulation in tumor samples.

The TPAC method is based on the evaluation of single samples by implementing Mahalanobis multivariate distance measurements on gene sets. The author claims that TPAC performs well both at population and sample-level. Gene sets definitions are inherited by MSigDB.

The calculate TPAC pathway scores by performing analyses on gene expression data for 21 solid human cancers from TCGA and associated normal tissues HPA.

TPAC seems to outperforms other single-sample enrichments methods, by providing better results with both patient prognosis and tumor stage.

The author also provide an R package ( https://hrfrost.host.dartmouth.edu/TPAC) that I did not test.

The method is clearly described, and the benchmarks show strong performances. However, a few issues with the approach and the manuscript writing have been identified and they are listed below:

Points required to be addressed.

Major

1. The main point of this paper seems to be the role of gene activity/expression as a proxy of how tumour cells translate genomic changes into functional advantages/malignancies, and the high cell-type specificity of gene expression, which calls for the development of tissue-specific and sample-specific enrichments. While figure 10 is remarkable in terms of how TPAC scores can be used to stratify tumour samples and eventually predict survival, the paper generally misses a more mechanistic discussion, especially in light of the MSigDB-centric view that is proposed. We understand that this paper scope is to describe a method, but once all this work is done, it would be nice to have a final figure with some reconstruction of the regulatory logic the TPAC score is underlying. Is there a set of transcription factors (TFs), besides MYC that explain higher scores? To answer this question, a very simple and immediate approach could be to apply decoupleR to the transcriptomes of samples with significant TPAC scores.

Do they converge towards a subset of meaningful TFs?

We are not bound to decoupleR and there are several other ways to answer this question, but following this tutorial would provide a potentially very interesting final figure to the paper in a very small amount of time.

https://www.bioconductor.org/packages/release/bioc/vignettes/decoupleR/inst/doc/tf_bk.html

2. One of the authors previous work is reported here as reference 11. In figure 1 of reference 11, there is a schematic in which "single-sample pathway enrichment" is reported. Also, Fig.2 of Ref 11 seems to provide similar information with respect to Fig. 6 and 8 of the current paper. We require the author to stress the differences between their previous work and the current manuscript, the advantages of the new method, ultimately comparing old and new results.

Minor

1) Line 13. We understand MSigDB has been used as a reference but there is a vast amount of gene set collections besides MSigDB. We see a value in sticking to a single collection, however, testing the efficacy of the method with other knowledge base would be demanding but not out of scope. Hence, we ask the author to at least mention alternatives to MSigDB that could be exploited when implementing TPAC.

2) Colours in Heatmaps don’t seem colour-blind friendly. This needs to be fixed.

3) The following sentence from the abstract “Because the null distribution of the TPAC scores has an accurate gamma approximation, both population and sample-level inference is supported.” is repeated equally at line 46 and 310. I understand it’s a bit of a mantra for the author but I’m quite sure it can be sacrificed or at least modified in the abstract or in the text.

**Have the authors made all data and (if applicable) computational code underlying the findings in their manuscript fully available?**

Reviewer #1: Yes

Reviewer #2: Yes

Reviewer #3: Yes

PLOS authors have the option to publish the peer review history of their article (what does this mean?). If published, this will include your full peer review and any attached files.

Reviewer #1: No

Reviewer #2: No

Reviewer #3: No
---

## [Decision Letter · Decision Letter 1]

27 Nov 2023

Dear Dr. Frost,

We are pleased to inform you that your manuscript 'Tissue-adjusted pathway analysis of cancer (TPAC): a novel approach for quantifying tumor-specific gene set dysregulation relative to normal tissue' has been provisionally accepted for publication in PLOS Computational Biology.

Best regards,

Martin H. Schaefer

Guest Editor

PLOS Computational Biology

Ilya Ioshikhes

Section Editor

PLOS Computational Biology

Reviewer's Responses to Questions

**Comments to the Authors:**

Reviewer #1: All comments have been addressed

Reviewer #3: The authors thoroughly answered all questions and criticisms raised, and I'm satisfied with the improvement made also in light of the other revisions.

**Have the authors made all data and (if applicable) computational code underlying the findings in their manuscript fully available?**

Reviewer #1: Yes

Reviewer #3: Yes

PLOS authors have the option to publish the peer review history of their article (what does this mean?). If published, this will include your full peer review and any attached files.

Reviewer #1: No

Reviewer #3: No

---

## [Editor Report · Acceptance letter]

3 Dec 2023

PCOMPBIOL-D-23-00735R1 

Tissue-adjusted pathway analysis of cancer (TPAC): a novel approach for quantifying tumor-specific gene set dysregulation relative to normal tissue

Dear Dr Frost,

I am pleased to inform you that your manuscript has been formally accepted for publication in PLOS Computational Biology. Your manuscript is now with our production department and you will be notified of the publication date in due course.

With kind regards,

Zsofia Freund
